# TRIM26 Facilitates HSV-2 Infection by Downregulating Antiviral Responses through the IRF3 Pathway

**DOI:** 10.3390/v13010070

**Published:** 2021-01-06

**Authors:** Tushar Dhawan, Muhammad Atif Zahoor, Nishant Heryani, Samuel Tekeste Workenhe, Aisha Nazli, Charu Kaushic

**Affiliations:** 1Department of Pathology and Molecular Medicine, McMaster University, Hamilton, ON L8S 4M3, Canada; dhawant@mcmaster.ca (T.D.); atif.zahoor@uhnresearch.ca (M.A.Z.); nishant.heryani@gmail.com (N.H.); nazlia@mcmaster.ca (A.N.); 2McMaster Immunology Research Center, Michael G. DeGroote Center for Learning and Discovery, McMaster University, Hamilton, ON L8S 4M3, Canada; 3Department of Pathobiology, Ontario Veterinary College, University of Guelph, Guelph, ON N1G 2W1, Canada; sworkenh@uoguelph.ca

**Keywords:** TRIM26, intrinsic immunity, interferon, ISG, HSV-2, epithelial cell, IRF3

## Abstract

Herpes simplex virus type 2 (HSV-2) is the primary cause of genital herpes which results in significant morbidity and mortality, especially in women, worldwide. HSV-2 is transmitted primarily through infection of epithelial cells at skin and mucosal surfaces. Our earlier work to examine interactions between HSV-2 and vaginal epithelial cells demonstrated that infection of the human vaginal epithelial cell line (VK2) with HSV-2 resulted in increased expression of TRIM26, a negative regulator of the Type I interferon pathway. Given that upregulation of TRIM26 could negatively affect anti-viral pathways, we decided to further study the role of TRIM26 in HSV-2 infection and replication. To do this, we designed and generated two cell lines derived from VK2s with TRIM26 overexpressed (OE) and knocked out (KO). Both, along with wildtype (WT) VK2, were infected with HSV-2 and viral titres were measured in supernatants 24 h later. Our results showed significantly enhanced virus production by TRIM26 OE cells, but very little replication in TRIM26 KO cells. We next examined interferon-β production and expression of two distinct interferon stimulated genes (ISGs), MX1 and ISG15, in all three cell lines, prior to and following HSV-2 infection. The absence of TRIM26 (KO) significantly upregulated interferon-β production at baseline and even further after HSV-2 infection. TRIM26 KO cells also showed significant increase in the expression of MX1 and ISG15 before and after HSV-2 infection. Immunofluorescent staining indicated that overexpression of TRIM26 substantially decreased the nuclear localization of IRF3, the primary mediator of ISG activation, before and after HSV-2 infection. Taken together, our data indicate that HSV-2 utilizes host factor TRIM26 to evade anti-viral response and thereby increase its replication in vaginal epithelial cells.

## 1. Introduction

Sexually transmitted infections (STI) constitute a significant global burden as they are responsible for significant morbidity and mortality, particularly in women, marginalized communities, and those who engage in high-risk sexual behavior. Herpes simplex virus type 2 (HSV-2), the virus responsible for genital herpes, is one of the most prevalent sexually transmitted infections [1]. According to the World Health Organization it is estimated that nearly 492 million people, aged 15–49 years, are living with HSV-2 worldwide [2]. Moreover, women constitute nearly 60 percent of infected individuals and the disease is reported to have the highest burden in Africa [2].

The epithelium that lines the female reproductive tract (FRT), is one of the primary physical barriers to pathogen entry and can be subdivided into three major compartments: lower, upper, and transitional. The lower compartment of the FRT, consisting of the vagina and the ectocervix, acts as the first line of defense of the mucosal immune system against STIs [3]. The lower FRT is lined with stratified squamous epithelial cells whereas the upper FRT is lined with a monolayer of columnar epithelial cells. Intercellular connecting structures, including tight junctions, adherents junctions, and desmosomal junctions, are responsible for maintaining mucosal barrier integrity [4].

Although the uterus provides support to the immunologically distinct fetus and is largely tolerant of allogeneic spermatozoa, the FRT is equipped with a variety of innate defenses to respond to a challenge from sexually transmitted pathogens [4,5]. These defenses consist of a variety of mechanical, chemical, and cellular factors. Notably, epithelial cells and the mucosal lining of the FRT constitute a physical barrier to pathogens. Secreted factors, including mucus and a variety of anti-microbial peptides, cytokines and chemokines, confer additional protection to the FRT [3]. Cellular factors consisting of a variety of innate immune cells, such as macrophages, dendritic cells, neutrophils, and natural killer cells, migrate to the FRT during the acute phase, in response to pathogenic exposure. Moreover, resident epithelial cells and stromal fibroblasts are also critical for host defense, playing a key role in mounting innate immune responses [3,5,6].

HSV-2 is classified to be part of the *Herpesviridae* family and encapsidates a double-stranded DNA genome [7]. It is transmitted through physical contact infecting epithelial cells at skin and mucosal surfaces, largely through asymptomatic or subclinical viral shedding during periods of reactivation. Following initial infection, the virus establishes latency in neural tissue, expressing certain latent-associated transcripts, specifically within the dorsal root or trigeminal ganglia [7].

Following initial viral exposure, the innate immune system mounts an antiviral response in order to restrict viral infection and replication, and to limit subsequent activation of the adaptive immune system [8]. The pattern recognition receptors (PRR) on host cells recognize viral signatures molecules, activating intracellular signaling cascades cumulating in significant decrease in viral load or even clearance of the virus. PRRs including toll-like receptors (TLRs), RIG-I-like receptors (RLRs), Nod-like receptors (NLRs) and DNA sensing systems which activate interferon regulatory factors (IRFs), recognize conserved pathogen-associated molecular pattern (PAMPs) of HSV-2 by various immune and non-immune cells [8,9]. Furthermore, viral infection triggers various molecular pathways downstream of these PRRs, which converge to recruit Tank-binding Kinase 1 (TBK1) and homologue IκB kinase epsilon (IKKε), to activate interferon regulatory factor 3 (IRF3) [10]. TBK1/IKKε acts as a complex which phosphorylates and activates transcription factors IRF3 and/or IRF7, resulting in type I interferon transcription [11,12]. To amplify type I IFN production, these interferons generate positive feedback loops by binding to the interferon-α/β receptor (IFNAR) through a paracrine and autocrine process. Consequently, this leads to the activation of the JAK/STAT pathway, resulting in the induction of various interferon-stimulated genes (ISGs) which initiate IFN-α transcription, thus amplifying type I interferon expression [13,14]. Type I interferons and their downstream ISGs activate crucial innate system components, as well host resistance to viral infections [13,14,15]. However, the specific mechanism of HSV-2 mediated innate immune activation in the urogenital system is poorly characterized. 

The tripartite motif (TRIM) protein family is composed of various ubiquitin E3 ligases which play an important role in regulating innate and intrinsic immunity [16,17]. For instance, it is well established that TRIM26, one of the proteins of the TRIM family, is characterized by the presence of three zinc-binding domains, RING domain, B-box type 1 region, B-box type 1 region, and coiled-coil region. However, research regarding the role of TRIM26 is limited [16]. A recent publication by Wang et al. provides considerable insight into the specific mechanisms through which TRIM26 exerts its effects [18]. The authors identified TRIM26 as an E3 ubiquitin ligase that promotes the K48-linked polyubiquitination and subsequent proteasomal degradation of phosphorylated IRF3. Specifically, they suggested that following viral infection, TRIM26 enters the nucleus in order to bind IRF3 leading to its degradation. Downstream factors such as IFN- β were also observed to be significantly decreased in TRIM26 transgenic mice following viral infection in vivo. Moreover, these mice were also more susceptible to infection. This suggests that viral pathogens may induce expression of TRIM26 as means of evading the innate immune system.

In previous studies, it has been shown that increased TRIM26 expression impaired the ability to inhibit replication of the RNA viruses Sendai virus (SeV) and Vesicular stomatitis virus (VSV) in vivo and in vitro [18]. Since there are very few studies that have examined the role of TRIM26 in suppressing anti-viral response, we decided to investigate the role of TRIM26 on HSV-2. We conducted a TRIM26 protein search and found a consensus dataset from The Human Protein Atlas suggesting that TRIM26 RNA is highly expressed in vaginal tissue [19,20]. We performed an analysis using RNA extracted from an immortalized vaginal epithelial cell line (VK2/E6E7) and found significant upregulation of TRIM26 after HSV-2 infection. Considering the role of TRIM26 in negatively regulating the antiviral response, we hypothesized that HSV-2 induces TRIM26 expression in order to attenuate innate immune responses.

To assess the influence of TRIM26 in negatively regulating the antiviral response of HSV-2 infection in vaginal epithelial cells, we performed functional studies on the underlying mechanism. We first created two cell lines derived from human vaginal epithelial cells (VK2) with TRIM26 overexpressed (OE) and TRIM26 knocked out (KO). When we infected all three cell lines, we found that high expression of TRIM26 was associated with increased HSV-2 infection and its absence results in reduced HSV-2 infection. We also found that TRIM26 negatively regulates the IRF3 signaling pathway in vaginal epithelial cells, thus affecting downstream IFN-β production and ISG activation.

## 2. Materials and Methods

### 2.1. Cell Line Maintenance

The WT VK2/E6E7 vaginal epithelial cell line was obtained from the ATCC (Manassas, VA, USA). This cell line (ATCC CRL-2616) was derived from normal human vaginal mucosal tissue and immortalized at passage 3 by transduction with the retroviral vector LXSN-16E6E7 in the presence of polybrene. The VK2 cells and the two TRIM26-modified VK2 variants (see below) were grown and maintained in keratinocyte serum-free medium (KSFM). KSFM was supplemented with an additional 0.1 ng/mL of human recombinant epidermal growth factor, 0.05 mg/mL of bovine pituitary extract, 0.4 mM CaCl_2_, and 100 units/mL penicillin-streptomycin (Life Technologies, Carlsbad, CA, US), as described before [21]. KSFM, human recombinant epidermal growth factor, and bovine pituitary extract were obtained from Life Technologies. Human embryonic kidney (HEK) 293T cells containing the SV40 T-antigen (ATCC Manassas) were cultured in high glucose formulation of Dulbecco’s modified eagle’s medium (DMEM) with 4.5 g/L of glucose and supplemented with 10% fetal bovine serum (FBS). African green monkey kidney or Vero cells (ATCC) were cultured in minimum essential medium eagle-alpha modification (α-MEM) supplemented with 5% of FBS and 1% of l-glutamine, penicillin-streptomycin, and HEPES.

### 2.2. Plasmid Constructs

Lentiviral plasmid expressing human TRIM26 expression was constructed using standard protocols by cloning the TRIM26 (NM_001242783.1) coding sequence into the pHR-CMV-ΔN-IκBα-IRES-Puromycin vector by replacing the ΔN-IκBα region with TRIM26 [22]. The resulting plasmid was designated as pHR-CMV-TRIM26-IRES-Puromycin. For CRISPR/Cas9-mediated knockout of TRIM26 gene, single-guide RNAs targeting TRIM26 were designed using CRISPR Design Tool (http://crispr.mit.edu/). Targeting sequence lies in the first exon of TRIM26. Oligo DNAs for the single-guide RNAs were annealed and inserted into lentiCRISPR v2 vector digested with BsmBI restriction enzyme as described [23]. All clones were confirmed by DNA sequencing using a primer 5′-GGACTATCATATGCTTACCG-3′ from the sequence of U6 promoter that drives expression of sgRNAs. Cassettes of single-guide RNAs-Cas9 or pHR-CMV-TRIM26 -IRES-Puromycin were introduced to VK2 cells by using lentivirus-mediated gene transfer.

### 2.3. Generation of Lentiviruses

Lentiviruses were produced in HEK 293T cells using the standard calcium phosphate method (Life Technologies) as described previously [22]. Briefly, 5 × 10^5^ cells were seeded onto 6-well plates in 2-mL medium one day before transfection. Cells were co-transfected on the next day when they had reached 70–80% confluency with lentiCRISPR v2 vector inserted with sgRNA or the lentiviral vector containing the TRIM26 transgene (1.5 ug), the packaging plasmid psPAX2 (1.0 ug), and the envelope plasmid VSV-G (0.5 ug). After 18 h (37 °C, 5% CO_2_), the medium was replaced with growth medium supplemented with 10% FBS, and at 24 h after the medium change, lentiviruses were harvested. The virus-containing media were centrifuged at 1500 rpm for 5 min, and the supernatant was collected and used to transduce VK2 cells. After 48 h of transduction, lentivirus transduced cells were selected using 1 μg/mL puromycin and depletion or over-expression of TRIM26 protein was confirmed.

### 2.4. Single Cell Clone Isolation

To isolate single cell clones of TRIM26 OE and KO cell lines, a limiting dilution technique was used as described previously [24]. Each transgenic cell line was prepared individually in a homogenized cell solution and diluted to a concentration of 10 cells/mL. 100µL of the diluted cell suspension was added into a 96-well plate and grown for 10 days in the presence of Puromycin-containing KSFM media for selective selection. Wells with single cells were marked and grown for 10–15 days in the presence of KSFM media containing Puromycin (1 µg/mL; Invivogen, San Diego, CA, USA) to populate each clone. The single cell clone wells were then expanded subsequently in flasks.

### 2.5. ALI Cultures and HSV-2 Infection

To generate an air-liquid interface (ALI) model, closely mimicking the physiological conditions of the lower FRT [21], a total of 60,000 cells from WT VK2, TRIM26 OE or TRIM26 KO were seeded on 0.4 µm pore-sized transwell polystyrene inserts (BD Falcon, Mississauga, ON, Canada) in standard KSFM growth medium (see above). After 24 h, media from the apical side was removed, whereas media in the basolateral side was replenished every 48 h for 7 days of culture ALI culture growth allows VK2 cells in vitro to grow in a multi-layered formation, thereby closely mimicking the stratified squamous epithelium analogous to the in vivo vaginal epithelium [21]. After 7 days of culture, VK2 cells were infected with wild-type HSV-2 (strain 333) at a multiplicity of infection (MOI) of 1, as previously described [21]. VK2 cells grown in ALI cultures survived with high viability up to 10 days and both their growth and level of HSV-2 infection was more dramatically altered by different hormonal treatments, when compared to standard single layer cultures [21]. The cells were then incubated at 37°C for 24 h and the supernatants were collected from the apical side for viral titration by plaque assay on Vero cells as described previously [21]. All VK2 transgenic cell lines were also infected with GFP-tagged HSV-2 for 2 h, washed with 1× PBS, incubated for 16 h and fixed with 4% paraformaldehyde. Cells were imaged using an EVOS™ FL digital inverted fluorescence microscope (Thermo Fisher Scientific, Burlington, ON, Canada).

### 2.6. Viral Titration

For plaque assays, Vero cells were grown in a monolayer to approximately 60% confluence in 12-well plates 1 day prior to infection. Culture supernatants were collected from VK2s grown in ALI cultures 24 h after infection with wild-type HSV-2, as described previously [21]. Supernatant samples were then serially diluted in serum-free α-MEM, added to the monolayer of Veros, and incubated for 2 h at 37 °C to facilitate viral absorption. The infected Veros were then overlaid with α-MEM containing 5% FBS and subsequently incubated for 48 h at 37 °C. After incubation, the cells were fixed and stained with crystal violet, and viral plaques were quantified using an inverted microscope. The number of PFU per milliliter was calculated by taking a plaque count for every sample and accounting for the dilution factor.

### 2.7. Lactate Dehydrogenase (LDH) Assay

WT VK2, TRIM26 OE, and TRIM26 KO cells were seeded for preparation of ALI cultures. After 24 h, the apical medium was removed to mimic ALI conditions. For each of the seven days after seeding the cells, medium was added to the apical side for 1 h and then collected for LDH assay. Collected samples were used to measure LDH using the assay kit according to manufacturer’s instructions (Pierce, Thermo Fisher Scientific, Waltham, MA, USA). LDH in culture supernatants was used as a marker of cell stress [25]. A positive control for maximum cellular LDH was used with supernatant collected after complete lysis of cells using lysis buffer supplied with LDH assay kit.

### 2.8. Cell Viability Assay

WT VK2, TRIM26 OE and TRIM26 KO cells were seeded at a cell density of 10,000 cells per well in 24 well plates. Every 24 h, a set of wells were trypsinized and the released cells counted with a Trypan blue exclusion assay by light microscopy in a hemocytometer. Cell growth and percent viability were recorded and used as measures of cell viability over time for 7 days. 

### 2.9. Immunofluorescent Staining and Confocal Microscopy

Immunofluorescent staining was performed as described previously [26]. Cells were briefly fixed in 4% paraformaldehyde and blocked for 30 min in blocking solution (5% Bovine serum albumin (Millipore-Sigma, Oakville, ON, Canada) and 5% normal goat serum (Millipore-Sigma) in 0.1% Triton X-100 (Bio-Rad Laboratories, Mississauga, ON, Canada) made in PBS). Primary antibodies mouse anti-TRIM26 (Santa Cruz Biotechnology, Dallas, TX, USA), mouse anti-total IRF3 (Santa Cruz Biotechnology), rabbit anti-phosho-IRF3 (Cell Signaling, Danvers, MA, USA), mouse IgG1κ isotype control (BD Pharmingen, San Diego, CA, USA) and rabbit IgG isotype control (Santa Cruz Biotechnology) were diluted in blocking solution in different combinations and incubated with cells for 1 h at room temperature. Following incubation with primary antibodies, the monolayers were washed with PBS and secondary antibodies, Alexa Fluor 488 goat anti-mouse IgG (Abcam, Cambridge, MA, USA) and Alexa Fluor 594 goat anti-rabbit (Invitrogen, Carlsbad, CA, USA) were added for 1 h at room temperature. After extensive washing, filters were excised from the polystyrene inserts and mounted on glass slides in mounting medium containing DAPI (Vector Labs, Burlingame, CA, USA). All samples were imaged on an inverted confocal laser-scanning microscope (Nikon Eclipse Ti2) using standard operating conditions (63× objective, optical laser thickness of 1µm, image dimension of 512 × 512, lasers: green 488 nm and red 594 nm laser lines). For each experiment, confocal microscope settings for image acquisition and processing were identical between control and treated cells and three separate, random images were acquired for analysis with each experimental condition. Image analysis was done by Image J software (NIH, Bethesda, MD, USA) to measure the areas of both fluorescently stained specific protein and cellular nuclei.

### 2.10. Quantitative Real-Time PCR

Human vaginal epithelial cells (VK2) were mock infected or infected with HSV-2 for 24 h at an MOI of 1 and total RNA was isolated using the RNAeasy kit (Qiagen, Toronto, ON, Canada). Relative quantitative real-time polymerase chain reaction (qRT-PCR) was performed using two-step SYBR green assays and the target genes were amplified with the following specific primers (forward and reverse primers, respectively): human *IFNα*: 5′-CTGGCACAAATGGGAAGAAT-3′ and 5′-CTTGAGCCTTCTGGAACTGG-3′; *IFNα5*: 5′-TGCTCAACTGCAAGTCAATCTGT-3′ and 5′-CTTGAGCCTTCTGGAACTGG-3′; IFN-λ1 5′-CACAGGAGCTAGCGAGCTTCA-3′ and 5′-TTTTCAGCTTGAGTGACTCTTCCA-3′; IFN-λ2/3 5′-GCCAAAGATGCCTTAGAAGAG-3′ and 5′-AGAACCTTCAGCGTCAGG-3′; IFN-γ primers 5′-TGCAGAGCCAAATTGTCTCC-3′and 5′-TGCTTTGCGTTGGACATTCA-3′; CCL2 5′-CCCCAGTCACCTGCTGTTAT-3′ and 5′-TGGAATCCTGAACCCACTTC-3′; IL-1β 5′-GGGCCTCAAGGAAAAGAATC-3′ and 5′-TTCTGCTTGAGAGGTGCTGA-3′; IL-8 5′-AGGGTTGCCAGATGCAATAC-3′ and 5′-CCTTGGCCTCAATTTTGCTA-3′; IL-11 5′-GAGACCTCCATTCAGGTGGA-3′ and 5′-TTGCAGTGACCTCAGATTGC-3′; TNFα 5′-ATCAGAGGGCCTGTACCTCA-3′; and 5′-GGAAGACCCCTCCCAGATAG-3′; IL-27 5′-GAGCAGCTCCCTGATGTTTC-3′ and 5′-AGCTGCATCCTCTCCATGTT-3′. The oligonucleotide primer sequences (Integrated DNA Technology IDT/MOBIX, Coralville, IA, USA) were used for interferon-stimulated genes (ISGs) and TRIM26 mRNA quantification. ISG specific gene primers were: *MX1* (accession number BC032602), Forward: 5′-CAGCACCTGATGGCCTATCA-3′, Reverse: 5′-ACGTCTGGAGCATGAAGAACTG-3′; and *ISG15* (accession number M13755), Forward: 5′-ACTCATCTTTGCCAGTACAGGAG-3′, Reverse: 5′-CAGCATCTTCACCGTCAGGTC-3′. *TRIM26* primers were: Forward: 5’-CTGGAAGTGCGTGACCTACA-3’; Reverse: 5′-CCCAGTCGTCATATCCATCC-3′. GAPDH primers (accession number NM_002046) were: Forward: 5′-ACAGTCAGCCGCATCTTCTTTTGC-3′; Reverse: 5′-TTGAGGTCAATGAAGGGGTC-3′. Quantitative real-time PCR was performed using the gene specific primer. The reaction was performed with RT^2^ SYBR^®^ Green qPCR Master mix according to the manufacturer’s manual (Qiagen) using the StepOne Plus™ Real-Time PCR System (Thermo Fisher, Waltham, MA, USA). Samples were run in triplicates and all data were normalized to GAPDH mRNA expression as an internal control. Fold change in gene expression was calculated in HSV-2 infected samples compared to mock uninfected controls.

### 2.11. IFN-β ELISA

Supernatants collected from VK2 cells after HSV-2 infection were analyzed for IFN-β production using human IFN-β ELISA kit (R&D Systems, Toronto, ON, Canada) according to manufacturer’s instructions. The minimum detectable limit of IFN-β ELISA kit ranges from 0.269–0.781 pg/mL and assay is specific to natural and recombinant human IFN-β.

### 2.12. Western Blotting

Whole-cell extracts were prepared by trypsinizing the VK2 cells and pelleting at 13,000× *g* for 15 min. Cell pellets were resuspended in RIPA buffer and lysed for 1 h on ice. Whole-cell lysates were clarified by centrifugation at 13,000× *g* for 10 min at 4 °C. Protein quantification was performed by BCA method as per manufacturer’s recommendations (Thermo Fisher, Waltham, MA, USA). Protein separation in polyacrylamide gels and immunoblotting were carried out as described previously [18]. Briefly cell lysates were resolved on 12% SDS–PAGE and electro-transferred to a nitrocellulose membrane (Amersham). After blocking with 5% non-fat milk in T-PBS overnight at 4 °C, the membranes were incubated with primary antibody: mouse monoclonal anti-TRIM26 (sc-393832; 1:5000 dilution) (Santa Cruz Biotechnology) at room temperature for 2 h. After washing, the membranes were incubated for 1 h with IRDye^®^ 680RD Donkey anti-Mouse IgG Secondary Antibody (1:15,000 dilution) (LI-COR, Lincoln, NE, USA). Expression of proteins was analyzed by LICOR-Odyssey infra-red scanner. Band intensities were quantified by ImageJ (NIH) software.

### 2.13. Plaque SIZE Measurements

Plaque size measurements were performed in 60–70% confluent monolayers of WT VK2, TRIM26 OE, TRIM26 KO cells in 24-well plates. Cells were then infected with HSV-2 strain 333 tagged with green fluorescent protein (GFP) for 2 h to allow virus adsorption. After 2 h, cells were washed three times with PBS to remove unbound viral particles and then overlaid with DMEM/F12 medium supplemented with 10% fetal bovine serum and 0.05% human immune serum incubated for 48 h at 37 °C. After incubation, the cells were fixed and viral plaques were quantified and imaged using a fluorescence microscope: EVOS™ FL digital inverted fluorescence microscope (Thermo Fisher, Waltham, MA, USA) at a ×40 magnification, and plaque size was measured using ImageJ software.

### 2.14. Statistical Analysis

Data presented are the mean ± SEM of representative data from multiple experiments generated using GraphPad Prism Version 8 (GraphPad Software, San Diego, CA, USA). Statistical significance was determined with two-tailed t-test between two treatments, one-way analysis of variance (ANOVA) between multiple treatments, and two-way ANOVA for the comparison of two different variables with their specific controls. where *p* < 0.05 was considered statistically significant. Multiple comparison p-values were adjusted with the Bonferroni post-hoc test. *p*-values for all respective analyses are indicated in the figure legends.

## 3. Results

### 3.1. HSV-2 Infection Upregulates TRIM26 Gene Expression in Vaginal Epithelial Cells (VK2s)

We first determined a time course for TRIM26 mRNA and protein expression in VK2 WT cells following HSV-2 infection. The VK2 cell line was chosen because it is an immortalized cell line from normal vaginal epithelial cells that can be genetically altered, passaged and grown easily in multilayers similar to squamous epithelium of the vaginal tissue allowing ALI culture, as described [21]. VK2 cells were grown in ALI cultures for 7 days and then infected with HSV-2 (strain 333) at an MOI of 1. RNA was extracted at 6, 12 and 24 h post HSV-2 infection for subsequent quantitative RT-qPCR to measure TRIM26 gene expression, compared to uninfected control cells. We found that HSV-2 infection upregulated TRIM26 gene expression 200-fold after 12 h of infection and 150-fold after 24 h post infection, as compared to mock controls (Figure 1A). In addition, we performed immunofluorescence staining for expression of the TRIM26 protein at the same time points. It was found that protein levels of TRIM26 increased significantly at 12 and 24 h post infection (Figure 1B,C). Interestingly, TRIM26 was found in the nucleus of HSV-2-infected VK2s but not in the mock control as seen in magnified images (Figure 1B). This suggests that infection with HSV-2 promotes accumulation of TRIM26 protein in the nucleus between 6 and 24 h after HSV-2 infection. This validates the previously reported effect of viral infection that promoted TRIM26 localization to the nucleus [18]. These observations indicate that HSV-2 infection upregulates TRIM26 mRNA and protein and promotes its localization to the nucleus in vaginal epithelial cells.

### 3.2. TRIM26 Facilitates HSV-2 Infection and Replication in VK2 Cells

Since TRIM proteins are known to play a role in antiviral host defense [27], and TRIM26 is known to be highly transcribed in human vaginal tissue [19,20], we hypothesized that TRIM26 plays a role in HSV-2 infection. To test our hypothesis, we first generated two transgenic cell lines from WT VK2 cells, one with overexpression (TRIM26 OE) of TRIM26 and the second with knocked out TRIM26 (TRIM26 KO), as described in materials and methods. Confocal immunofluorescence microscopy and RT-qPCR analysis were used to validate the altered expression of TRIM26 in transgenic TRIM26 OE and TRIM26 KO cell lines. The results indicated that TRIM26 mRNA (measured by qPCR) and protein (measured by immunofluorescence and Western blot) are highly expressed in TRIM26 OE cells compared to WT VK2 cells (Figure 2A–D). In addition, overexpression of TRIM26 in VK2 cells had no effect on cell growth and no measurable change in viability, nor indication of metabolic stress when compared to WT VK2, as assessed by viable cell counts and LDH release assays over 7 days of culture (Figure 2E,F). The TRIM26 KO cells demonstrated nearly absent TRIM26 mRNA and intracellular TRIM26 protein compared to WT VK2 cells (Figure 3A–D). Knocking out TRIM26 in VK2 cells had no effect on cell growth or viability and did not show any increase in LDH release, as compared to WT VK2 cells over 7 days of culture (Figure 3E,F).

With the successful derivation and validation of both TRIM26 OE and KO VK2 cell lines, we performed functional studies to further elucidate the role of TRIM26 in HSV-2 infection. To see whether there is any possible effect of TRIM26 on initial uptake of HSV-2 we infected all three cell lines WT VK2, TRIM26 OE and TRIM26 KO cell lines for 2 h and 16 h after which cells were fixed and stained for intracellular HSV-2. Images were captured as Z-stacks by immunofluorescence microscope and every slice of Z-stack was quantified for viral fluorescence by Image J software to predict viral load in all three cell lines (Figure 4A,B). There was no difference in the amount of virus initially internalized by the three cell lines at the 2 h time point, but after 16 h the viral load in TRIM26 OE far exceeded that of VK WT and TRIM26 KO, the latter showing baseline immunofluorescence. Viral shedding was quantified after HSV-2 infection (MOI = 1) in both OE and KO cell lines, relative to WT VK2 cells, as an indicator of viral replication. All three cell lines were infected with HSV-2 for 24 h and supernatants were collected for viral titration using the Vero cell plaque assay. We observed that overexpression of TRIM26 increased HSV-2 shedding significantly (~10-fold higher) into the supernatants relative to WT VK2 cells, whereas a significantly lower amount of virus was found in supernatants of HSV-2-infected TRIM26 KO cells (~100-fold lower) compared to TRIM26 OE cells and WT VK2 cells (Figure 4C).

We also analyzed various cytokines and chemokines gene expression in WT VK2, TRIM26 OE and TRIM26 KO cell lines to see if the genetic manipulation of these cell lines has any off-target effect. The results are shown in Appendix A. We found that other than Interleukin-8 (IL-8) and CCL2 also known as monocyte chemoattractant protein 1 (MCP1) there was no significant difference in any other cytokine and chemokine gene expression of these cell lines.

Cells from all three cell lines were grown individually in monolayers and infected (MOI = 1) with HSV-2 tagged with green fluorescent protein (HSV-2-GFP). Cells were fixed 16 h post infection, and images were captured by fluorescence microscopy using an inverted fluorescence microscope (EVOS cell imaging systems, Thermo Fisher Scientific). HSV-2-GFP infected cells appear bright green (Figure 4D). Results indicated that there was an increase of HSV-2-GFP expression in the TRIM26 OE cell line with high levels of virus compared to WT control (Figure 4B). In contrast, the TRIM26 KO cell line showed very few infected cells relative to WT control. Furthermore, TRIM26 OE cells had significant cytopathic effects as seen by large clear plaques of lysed cells, indicating high virus production, while TRIM26 KO cells showed no cytopathic effects. The average plaque size produced by HSV-2 infection of the three cell lines revealed that the TRIM26 OE cell cultures had the largest plaques, approximately 2-fold larger that WT controls, and the TRIM26 KO cell cultures had much smaller plaque size compared to both of the other cell cultures.

Thus, the viral titre and immunofluorescence results indicated that increase of TRIM26 expression led to enhancement of HSV-2 replication and the absence of TRIM26 caused greatly reduced HSV-2 replication of vaginal epithelial cells. These data suggest that TRIM26 upregulation by HSV-2 plays a crucial role in augmenting HSV-2 infection.

### 3.3. TRIM26 Expression Negatively Regulates IRF3

Next, we examined the mechanism by which TRIM26 was enhancing HSV-2 replication. Previous studies reported that TRIM26 targets IRF3, one of the main transcription factors involved in production of IFN-β, resulting in IRF3 ubiquitination and subsequent proteasomal degradation [18]. Normally, IRF3 is phosphorylated (pIRF3) and translocated to the nucleus in this activated state [28]. To investigate the role of TRIM26 on IRF3 activation in the context of HSV-2 infection, immunofluorescence staining was performed following HSV-2 infection of WT, TRIM26 KO and TRIM26 OE cells to examine the localization of IRF3 and pIRF3, using confocal microscopy. Positive controls were performed, where all 3 cell types were activated by poly (I:C), a known stimulator of IFN-β through the IRF3 pathway [29]. After 1 h of HSV-2 infection, all infected, uninfected and poly(I:C) treated cells were fixed and stained with fluorescent antibodies against total IRF3 (green fluorescence) and pIRF3 (red fluorescence). Uninfected TRIM26 KO cells displayed significantly higher fluorescence for both total IRF3 and pIRF3, primarily localized in the nucleus (Figure 5). This would be consistent with a dysregulation of IRF3 activity leading to an increase in pIRF3 in the absence of TRIM26. While low levels of IRF3 and pIRF3 were present in the nuclei of WT cells, IRF3 was nearly absent from the nuclei and cytoplasm of TRIM26 OE cells. This indicates that over-expression of TRIM26 leads to substantial reduction in IFR3, while in normal cells modest baseline IRF3 activity is seen in the absence of infection. After 1 h of HSV-2 infection, WT cells had substantial cytoplasmic IRF3 and nuclear pIRF3, while TRIM26 OE had substantial cytoplasmic IRF3, but very little nuclear pIRF3. In contrast TRIM26 KO cells had very high levels of nuclear pIRF3. Infection of WT cells leads to IRF3 activation and pIRF3 location in the nucleus as expected, but this does not occur in TRIM26 OE cells and is exaggerated in the response of TRIM26 KO cells. In addition, we also analyzed all three cell lines at a later time point (19 h) after HSV-2 infection and found that maximum IRF3 activation occurred within 30 min compared to 2 h after HSV-2 infection. At later time points post-infection (19 h), minimal to no IRF3 expression was observed (Appendix A). When the three cell types were treated with poly (I:C), TRIM26 KO cells show significantly higher accumulation of pIRF3 to the nucleus compared to the WT VK2 cells, consistent with translocation into the nucleus upon activation, whereas TRIM26 OE cells showed IRF3 accumulation in the cytoplasm with very little pIRF3 in the nucleus. Thus, the positive controls paralleled the differential response of the cell types to HSV-2 infection. These results confirm that increased TRIM26 likely targets pIRF3 for degradation in the nucleus and in the absence of TRIM26, substantial pIRF3 accumulates in the nucleus in the context of HSV-2 infection. These data support the previous finding that TRIM26 plays a role in the nuclear degradation of IRF3 [18] and demonstrates its role in the context of HSV-2 infection.

### 3.4. TRIM26 Negatively Regulates Expression of IFN-β and Interferon Stimulated Genes (ISGs) through IRF3 Pathway during HSV-2 Infection

To confirm that HSV-2 infection is controlled through the IRF3 signaling pathway under the influence of TRIM26, WT VK2 and TRIM26 KO cell lines were treated with BX-795, an inhibitor of TBK1 and IKKɛ which are upstream of IRF3 signaling [30]. Cells were treated with BX-795 for an hour after which the inhibitor was removed and cells were subsequently infected with HSV-2-GFP (MOI = 1) (Figure 6A). Both WT VK2 and TRIM26 KO cell lines showed significant increase in HSV-2 GFP infection after BX-795 treatment, a result that confirms that inhibition of the IRF3 pathways contributes to increased HSV-2 infection.

Since TRIM26 upregulation was seen to directly downregulate pIRF3, we next examined anti-viral factors downstream from IRF3. Current evidence suggests that expression of type I interferons can be induced through recognition of cytosolic DNA, such as the genome of herpes simplex virus post infection, in a process known as DNA sensing [31]. Thus, we examined IFN-β expression after HSV-2 infection, and the expression of the IRF3-dependent ISGs, MX1 and ISG15. All three cell lines were grown in ALI conditions, infected with mock control or HSV-2 (MOI = 1). IFN-β was first analyzed by qPCR following 6 h and 24 h of HSV-2 infection. TRIM26 KO cells show significantly higher IFN-β gene expression following 6 h of HSV-2 infection, whereas TRIM26 OE cells significantly inhibit IFN-β expression relative to WT VK2s (Figure 6B). Interestingly, only TRIM26 KO cells showed significant increase in IFN-β gene expression 6 h after HSV-2 infection relative to its mock conditions, suggesting that knockout of TRIM26 allows for a more potent IFN-β response after HSV-2 infection. Supernatants were also collected prior to and following 24 h of HSV-2 infection for the quantification of IFN-β production with a commercial IFN-β ELISA kit. We found that the TRIM26 KO cells showed enhanced production of IFN-β post HSV-2 infection compared to WT VK2 cells (Figure 6C). In fact, IFN-β production by uninfected TRIM26 KO cells was significantly higher than uninfected WT VK2 cells. However, TRIM26 OE cells showed no increase of IFN-β after infection compared to WT VK2 cells. TRIM26 KO cells showed consistency in the increase in IFN-β gene expression and IFN-β protein production between mock and HSV-2-infected conditions. These data clearly indicate that the absence of TRIM26 leads to elevation of IFN-β production.

The IRF3-dependent ISGs MX1 and ISG15 were analyzed by qPCR following 6 h of HSV-2 infection. Uninfected control TRIM26 KO cells showed small but significant increased baseline expression of MX1 and ISG15, whereas the expression of these ISGs was not altered in TRIM26 OE cells, as compared to WT VK2s (Figure 6D,E). Following HSV-2 infection, both TRIM26 KO and WT VK2 cell lines showed significant increase in MX1 and ISG15 expression relative to their uninfected controls, whereas TRIM26 OE showed no significant difference in ISG expression after infection (Figure 6D,E). These data clearly indicate that the absence of TRIM26 leads to elevation of IFN- β and ISGs by 6 h post infection.

Although TRIM26 has a direct link with IRF3 to allow for indirect modulation of ISG production, we further investigated whether ISG production is IFN-β dependent. Given that IFN lambda 1–3 are known to upregulate MX1 and ISG15, we performed qPCR for expression of these genes along with other interferons before and after 6 and 24 h of HSV-2 infection. Interestingly, we found significantly higher gene expression of IFNα, IFNα5, IFN-γ, IFN-λ1 and IFN-λ2/3 in TRIM26 KO cells 6 h after HSV-2 infection relative to mock conditions (Appendix A). This suggests that production of MX1 and ISG15, and possibly other antiviral factors, may also be connected through complex mechanisms in addition to IRF3 modulation by TRIM26. This hypothesis will need further investigation.

Overall, these results suggest that TRIM26 acts as a negative regulator of IRF3 signaling pathway through the ISGs such as MX1 and ISG15, thereby contributing to the attenuation of antiviral responses against HSV-2.

## 4. Discussion

The previous literature has shown that overexpression of TRIM26 increases viral replication, such as the RNA virus VSV [18]. Using TRIM26 transgenic cell lines, we demonstrated that HSV-2 infection is also altered with changes in expression of TRIM26. We have shown that TRIM26 serves as a regulator of the antiviral response to HSV-2 by demonstrating that modulation of TRIM26 expression alters the activity of IRF3 and the outcome of HSV-2 replication (Figure 7B). Overexpression of TRIM26 significantly increased viral shedding, whereas knockout of TRIM26 resulted in little release of HSV-2, compared to WT VK2 cells. These initial data strongly suggested that TRIM26 may play an important and undiscovered role in the context of HSV-2 infection of the vaginal epithelium. A simplified illustration of the probable mechanism through which TRIM26 acts in response to HSV-2 infection in vaginal epithelial cells (VK2) is shown in Figure 7A. Our results are in agreement with the results of the study done by Wang et al. [18], as we have confirmed the role of TRIM26 as a negative regulator of anti-viral response, extending this conclusion to HSV-2 infection in vaginal epithelial cells. To the best of our knowledge, our study is the first to report the increased expression of a TRIM protein by HSV-2 as a mechanism to increase its replication in host cells.

The establishment of a productive viral infection is dependent on the net outcome from the contest between the anti-viral innate responses mounted by the host cell and the immune evasion ability of the virus. While being the target of various cellular proteins that inhibit replication, HSV-2 is also able to activate many proteins expressed by host cells in order to achieve viral propagation. HSV-2 has a number of viral proteins that interfere with viral sensing and various immune signaling pathways to enhance a successful viral infection [32,33]. For instance, herpesvirus viral protein US3 is known to activate Akt signaling to promote the phosphorylation of cGAS, a protein involved in viral dsDNA sensing. US3 also blocks activation and translocation of IRF3 to the nucleus, preventing a type I interferon response against HSV-2 [32]. An alternate to a viral protein mechanism is the ability of the virus to upregulate host proteins that subvert the innate immunity. The recent literature suggests that members of the TRIM protein family play a critical role in the regulation of innate immune responses and antiviral immunity [16,34]. However, there are very few papers investigating the role of TRIM proteins in HSV infection, and none that identify or determine a role for TRIM26 in epithelial cells. TRIM5α has been shown to restrict both HSV-1 and HSV-2 infection when transfected into HeLa cells [35]. Another study using fibroblast cell lines identified TRIM43 as upregulated in HSV-1 infection and indicated a role for TRIM43 in controlling viral lytic gene expression by regulation of host centrosomes and the integrity of nuclear lamina [36]. Two other studies identified potential roles for TRIM9 and TRIM14 in the induction of type I IFN responses after HSV-1 infection, using in vitro studies with non-epithelial cell lines [37,38]. Each of the TRIM proteins in those studies has very low or undetectable expression in the FRT when compared to the high expression of TRIM26 [19,20]. Other TRIM proteins including TRIM5α and TRIM43, unlike TRIM26, have direct anti-viral roles. Our results show that given the abundant expression of TRIM26 in vaginal epithelial cells, this protein is likely being used by HSV-2 to manipulate the host innate immunity.

Using immunofluorescent microscopy, we showed that infection with HSV-2 induces nuclear accumulation of TRIM26. TRIM26 has been established as an E3 ligase that promotes nuclear IRF3 ubiquitination and its subsequent degradation [18]. Moreover, it has been shown that TRIM26 interacts with IRF3 in the nucleus after infection with Sendai virus (SeV) in macrophages [18]. We demonstrated by immunofluorescence staining and confocal microscopy, that overexpression of TRIM26 attenuated phosphorylated IRF3 expression despite an activation of IRF3 response by HSV-2 infection and poly (I:C) treatment, while TRIM26 KO cells displayed increased expression of phosphorylated IRF3. This indicates that in VK2 cells, TRIM26 targets phosphorylated IRF3 for degradation in the nucleus. Therefore, TRIM26 appears to play a role in mediating antiviral innate immune responses against HSV-2 infection in VK2 cells.

To investigate the impact of TRIM26 expression on innate immune pathways downstream of IRF3, we decided to examine production of IFN-β and ISGs. One previous study demonstrated that low dose overexpression of TRIM26 was able to potentiate SeV-induced IFN-β promoter activation in HeLa cells, but not in response to HSV-1 infection, suggesting that the role of TRIM26 specifically could be different for RNA viruses compared a DNA virus [39]. In our results, HSV-2, which is a DNA virus, showed that TRIM26 KO cells had significantly higher production of IFN-β and the antiviral ISGs, MX1 and ISG15, after HSV-2 infection, compared to WT and TRIM26 OE cells. It is important to note that TRIM26 KO cells show an increase in IFN-β production after 24 h of viral infection, whereas the expression of ISGs increased at an earlier stage of 6 h post infection. Literature suggests that several ISGs, including MX1 and ISG15, have been found to upregulated after human cytomegalovirus (HCMV) infection and behave in an interferon-independent, IRF3-dependent manner [40]. Although we have not confirmed this in our in vitro model, it is likely that we may be seeing a similar behavior of these ISGs in the context of HSV-2 infection of vaginal epithelial cells. Interestingly, analysis of ISG expression indicates that TRIM26 OE cells had no significant increase in MX1 and ISG15 expression after infection, in contrast to WT VK2 and TRIM26 KO cells, suggesting that TRIM26 may have a direct regulatory role for these ISGs. Interestingly, other IFNs such as IFN-λ and IFN-α have been known to upregulate ISGs such as MX1 and ISG15 [41,42]. One study showed that ISG15 regulates IFNγ immunity, which was consistent with the increased expression of IFNγ that we saw as well [43]. This suggests that TRIM26 may have important implications beyond type I interferons which are not under the regulation of IRF3, however; their complex signaling behavior requires further investigation.

Indeed, over-expression of TRIM26 had no effect on IFN-β production in response to HSV-2. The differences in results between our work with HSV-2 and Ran et al. with SeV, may well be the result of the differences in response to DNA versus RNA viruses, but may also represent differences in the target cell type as all of our experiments were performed in vaginal epithelial cells while their work was in HeLa cells. Interestingly, a separate study found that ISG15 expression provides enhanced innate antiviral response by stabilizing IRF3 [44]. Therefore, this may indicate that knocking out TRIM26 allowed for the enhancement of an antiviral IFN-β and ISG positive feedback response in the context of HSV-2 infection. Further studies will need to be conducted to elucidate any differences in Type 1 IFN response mechanisms to RNA versus DNA viruses that related to the role of TRIM26.

Although our study shows the clear role of TRIM26 in modulating specific antiviral responses, it is important to acknowledge the complexity of various HSV-2 proteins that play similar roles in suppressing interferon signaling, and thus allowing replication. For example, ICP27 has been shown to suppress IRF3 activation and downstream IFN-β production in epithelial cells [45]. Other HSV proteins such as ICP0, ICP22 and vhs have also been shown to exhibit similar effects [46,47]. It is clear that HSV-2 suppresses the host antiviral response through the expression of its own proteins along with the modulation of various host proteins, such as TRIM26, that play a role in regulating cellular intrinsic and innate immune signaling pathways. The characterization of viral and host protein interactions has not been studied extensively and needs to be examined more in-depth. It would be interesting to further investigate whether and which HSV proteins interact with TRIM26 and other TRIM proteins and determine if those interactions alter infection and intrinsic immunity.

A growing body of evidence identifies HSV-2 infection as a risk factor for acquiring HIV [48]. The outcome of viral exposure in the female genital mucosa is determined by interaction of the virus with various innate anti-viral factors, including those induced by the epithelial cells lining the mucosal surface [49]. An inflammatory microenvironment can increase the risk of HIV-1 acquisition by weakening the barrier on mucosal surfaces and through recruitment of target cells that allow establishment of productive infection [50]. Other research has linked HSV-2 infection with increased susceptibility to HIV-1, because HSV-2 increases proinflammatory cytokines and chemokines that facilitate recruitment of HIV target cells to the mucosal barrier [51,52]. In previous studies, we have shown that exposure to HIV-1 can breach the mucosal barrier because of the production of proinflammatory cytokines, mainly TNFα, by genital epithelial cells [26]. However, HIV-1 exposure also induced IFN-β in genital epithelial cells which opposes the effect of TNFα and plays a protective role against epithelial barrier disruption [53]. In the current study we showed that HSV-2 can potentially downregulate IFN-β production through upregulation of TRIM26-mediated IRF3 degradation. We would therefore posit that the downregulation of INF-β by HSV-2 could also contribute to increased susceptibility of the genital mucosa to HIV-1 infection.

Our study poses some potential limitations that should be considered in future studies. Firstly, although the use of well-characterized immortalized VK2 cell line aids in providing reproducible results, these cells do not represent the biological variation among normal vaginal tissue cells in women. Furthermore, despite ALI culture conditions being strong representations of vaginal epithelium morphology, this in vitro model does not completely characterize the complex microenvironment of the vaginal tract. The vaginal mucosa of the lower reproductive tract is covered with mucus containing mucin, electrolytes and proteins such as lysozymes, immunoglobulins and lactoferrins, which all act through unique mechanisms to defend against pathogens [41]. Additionally, the vaginal epithelium is under the influence of female sex hormones, which vary with the menstrual cycle. Female sex hormones, such as estrogen and progesterone, have been known to impose effects on epithelial cells, fibroblasts and immune cells in the FRT to alter their functions and therefore susceptibility to STIs [4,5,42]. Therefore, for clinical application, this will need to be confirmed with in vivo or clinical studies.

In summary, our work has unveiled a mechanism by which HSV-2 is able to hijack a host protein, TRIM26, in order to attenuate innate antiviral responses and thereby benefit its infection and replication in vaginal epithelial cells of the FRT. Given the antiviral role of IFN-β and ISGs in providing protection against HSV-2, TRIM26 could be a useful target for potential therapeutic drugs to limit HSV-2 infection, particularly in women.

## Figures and Tables

**Figure 1 viruses-13-00070-f001:**
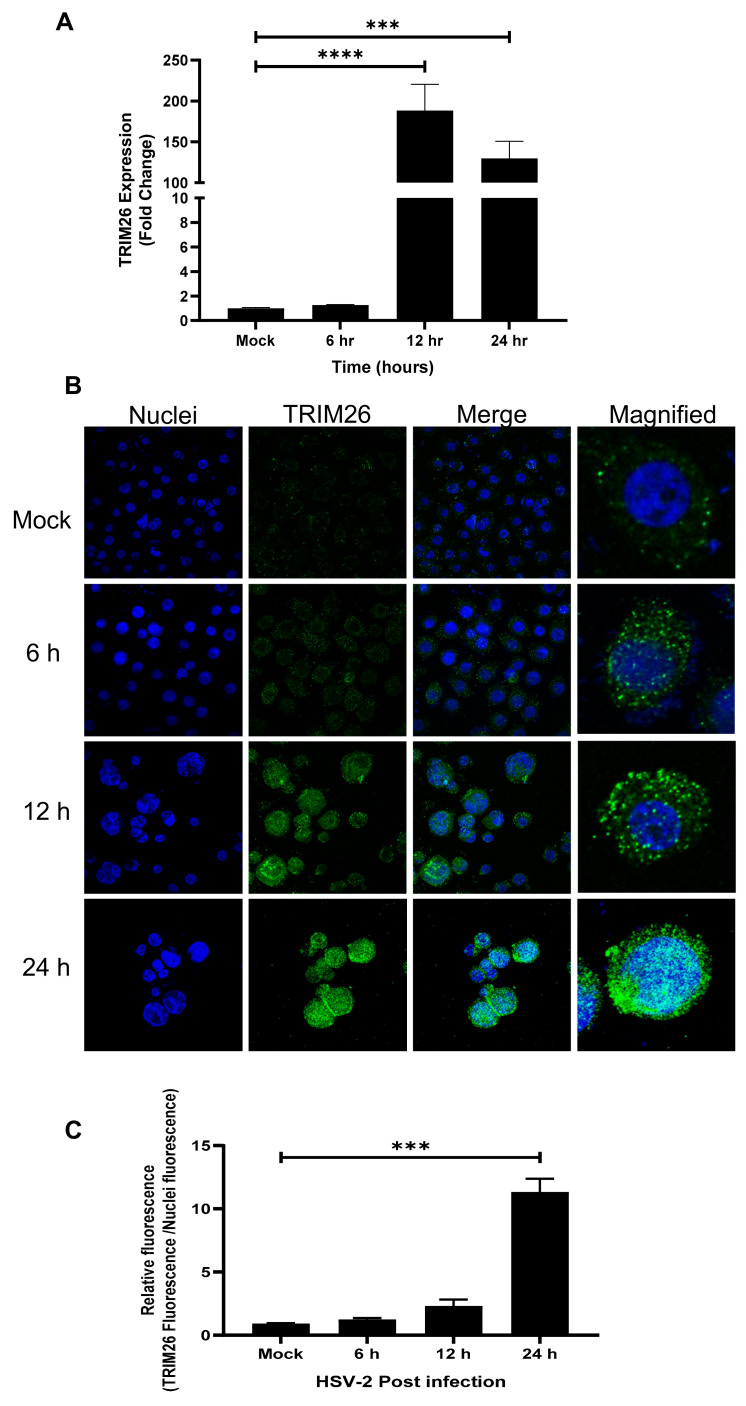
Herpes simplex virus 2 (HSV-2) infection upregulates TRIM26 expression in vaginal epithelial (VK2) cells. VK2s grown in ALI cultures were either uninfected (mock) or infected with HSV-2 (Multiplicity of Infection = 1) and RNA was extracted from the cells or cells were fixed for staining at 6, 12 and 24 h post HSV-2 infection. (**A**) qPCR was performed to measure TRIM26 expression and results shown as fold change compared to non-infected controls. Data show mean fold change +/− SEM (*n* = 6); statistical significance *** *p* = 0.0001; **** *p* < 0.0001. (**B**) Representative immunofluorescence images of VK2 cells showing TRIM26 (green) protein and nuclei (blue) staining, by confocal microscopy. Mock infected control cells are compared to infected cells (6, 12, 24 h post infection). Merged image (third column) contains both blue from nuclei and green from TRIM26 protein (Magnification x630). Magnified images of single cells are shown (right column). (**C**) Relative fluorescence of TRIM26 vs. nuclear stain was quantified using ImageJ software. Data shown are mean + SEM. Statistical significance relative to mock control *** *p* = 0.0008.

**Figure 2 viruses-13-00070-f002:**
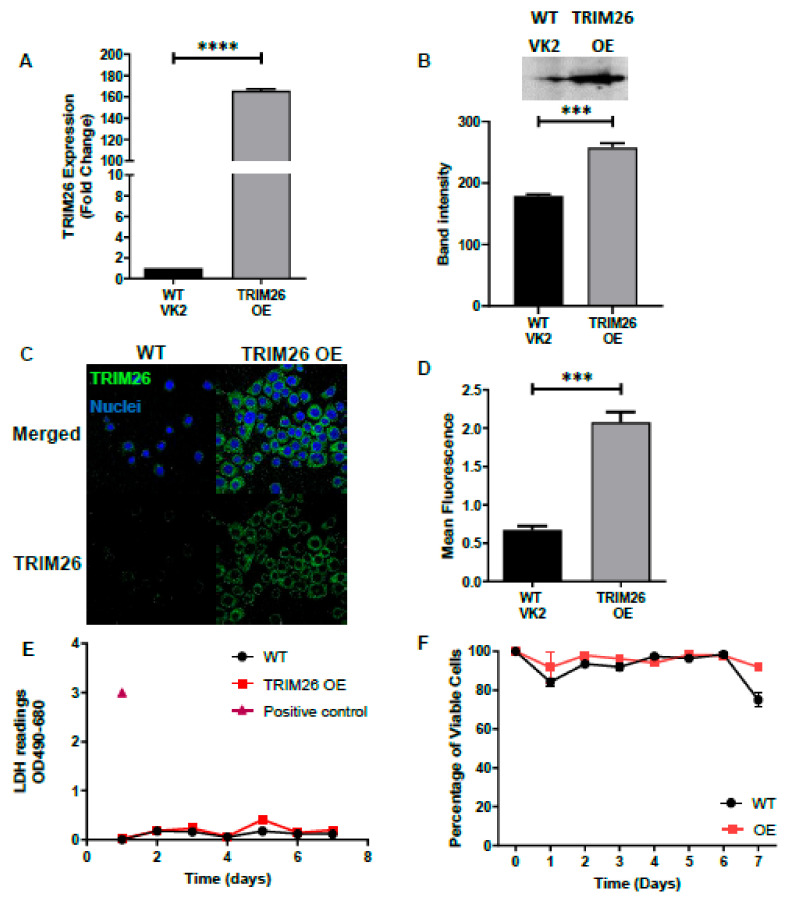
Validation of TRIM26 over expression (OE) cell line. TRIM26 OE cell line was validated by comparison to wild type (WT)VK2 cell line for TRIM26 mRNA and protein expression levels. (**A**) TRIM26 gene expression in WT VK2 and TRIM26 OE cell lines were analyzed by RT qPCR as fold change relative to uninfected WT controls. Results show mean ± SEM fold change (*n* = 4); Statistical significance: **** *p* < 0.0001. (**B**) TRIM26 protein expression analyzed in WT and TRIM26 OE cell lysates by Western blot. Band intensity was measured by ImageJ software for *n* = 3 samples; statistical significance: *** *p* < 0.001 (**C**) Representative fluorescent images (*n* = 5) with immunofluorescence staining of TRIM26 (green) and nuclear DNA stain (blue) for WT VK2 and TRIM26 OE, by confocal microscopy. Magnification x630. (**D**) Fluorescence of TRIM26 protein from immunofluorescent images was measured by Image J software and mean + SEM calculated (*n* = 5). Statistical significance: *** *p* < 0.001. Supernatant collected from cell lines was used for lactate dehydrogenase (LDH) assay (**E**) and to quantify percent viable cells by Trypan blue dye exclusion test (**F**) comparing WT VK2 with TRIM26 OE cells during 7 days of culture. Positive control total cell LDH shown in (**E**). Data are mean ± SEM for (*n* = 3) replicate cultures for both (**E**) and (**F**).

**Figure 3 viruses-13-00070-f003:**
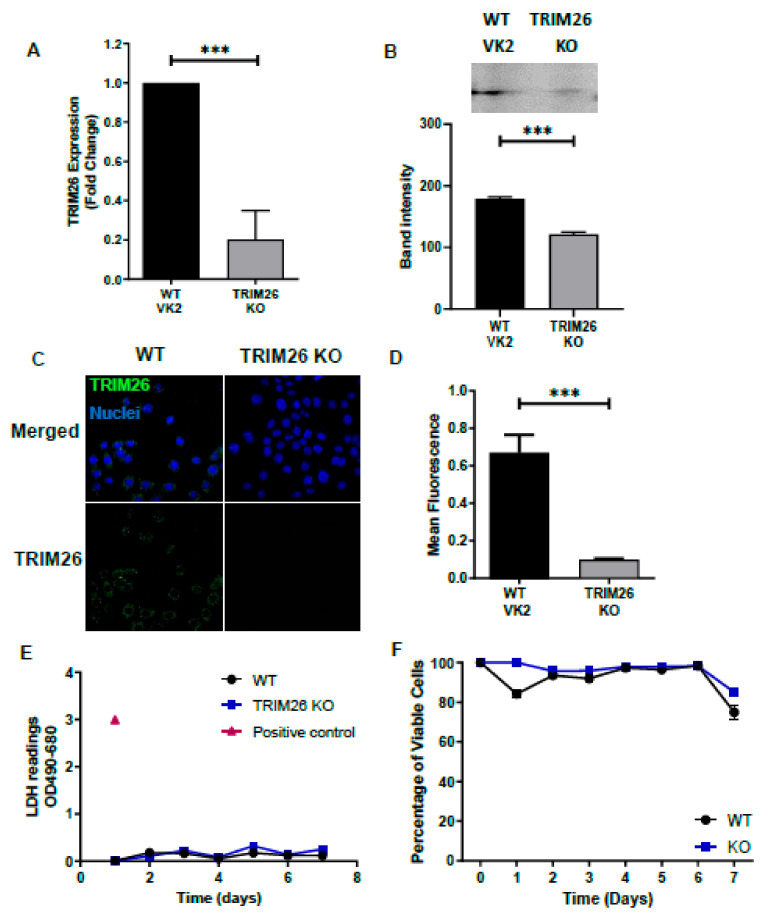
Validation of TRIM26 Knock out (KO) cell line. TRIM26 OE cell line was validated by comparison to WT VK2 cell line for TRIM26 mRNA and protein expression levels. (**A**) TRIM26 gene expression in WT VK2 and TRIM26 KO cell lines were analyzed by RT qPCR as fold change relative to uninfected WT controls. Results show mean ± SEM fold change (*n* = 4); Statistical significance: *** *p* < 0.0003. (**B**) TRIM26 protein expression analyzed in WT and TRIM26 KO cell lysates by Western blot. Band intensity was measured by ImageJ software for *n* = 3 samples; statistical significance: *** *p* < 0.001. (**C**) Representative fluorescent images (*n* = 5) with immunofluorescence staining of TRIM26 (green) and nuclear DNA stain (blue) for WT VK2 and TRIM26 KO by confocal microscopy. Magnification x630. (**D**) Fluorescence of TRIM26 protein from immunofluorescent images was measured by Image J software (*n* = 5) and mean + SEM calculated. Statistical significance: *** *p* < 0.0003. Supernatant LDH cell viability test (**E**) and percent viable cells by Trypan blue dye exclusion (**F**) comparing WT VK2 with TRIM26 OE cells during 7 days of culture. Positive control total cell LDH shown in (**E**). Data are mean ± SEM for (*n* = 3) replicate cultures for both (**E**) and (**F**).

**Figure 4 viruses-13-00070-f004:**
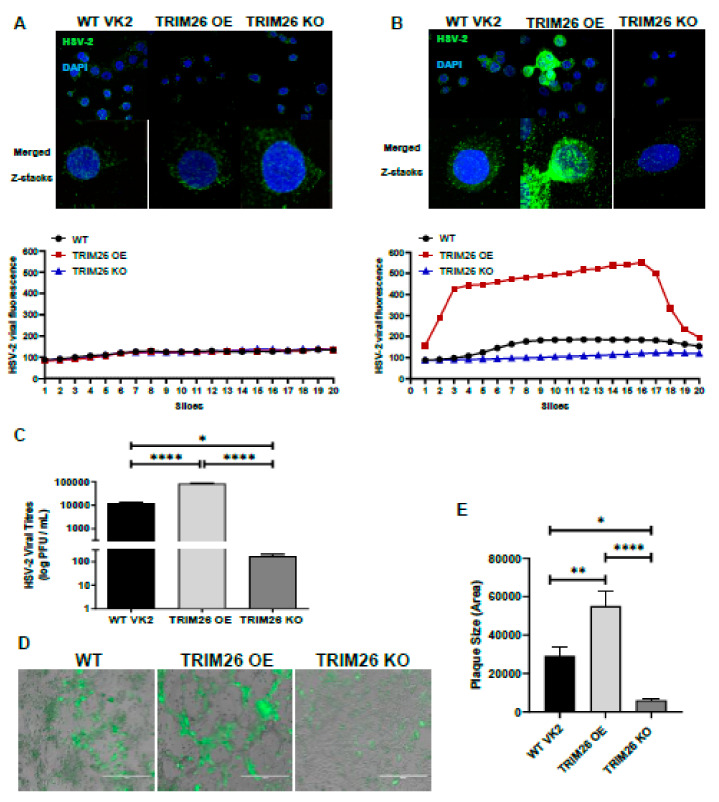
TRIM26 over expression enhances HSV-2 infection of VK2 cells. All three cell lines were exposed to HSV-2 for 2 h at 4 °C and then incubated for (**A**) 2 h and (**B**) 16 h at 37 °C. The cells were fixed and intracellular HSV-2 stained by immunofluorescence (green) and nuclei stained with DAPI (blue), with visualization by confocal microscopy. Top images are views of multiple cells (x630), while bottom images are single cells (x2520) with merge of all signals through the cells (all Z-stacks). Charts below indicate HSV-2 fluorescence (*y*-axis) quantitated for each Z-stack slice (*x*-axis) from representative single cells above. (**C**) WT VK2, TRIM26 OE and TRIM26 KO cell lines were infected with HSV-2 (MOI = 1) for 24 h. HSV-2 titres were performed using supernatants collected 24 h post infection. Data shown as mean ± SEM (*n* = 5). Statistical significance: * *p* = 0.0002; **** *p* < 0.0001. (**D**) All three cell lines were infected with HSV2-GFP (MOI = 1) for 16 h before fixation. Cells were visualized using an EVOS fluorescent microscope under low magnification (10 × objective). Representative images are shown for each cell type. Higher levels of HSV-GFP fluorescence are evident in TRIM26 OE cells relative to WT and KO cells with a high degree of cell aggregation indicative of plaque formation. White line indicates 400 μm measurement. (**E**) Plaque size was measured by ImageJ software and plotted as bar graph. Data indicate mean ± SEM (*n* = 6) with statistical significance: * *p* = 0.02, ** *p* = 0.008, **** *p* < 0.0001.

**Figure 5 viruses-13-00070-f005:**
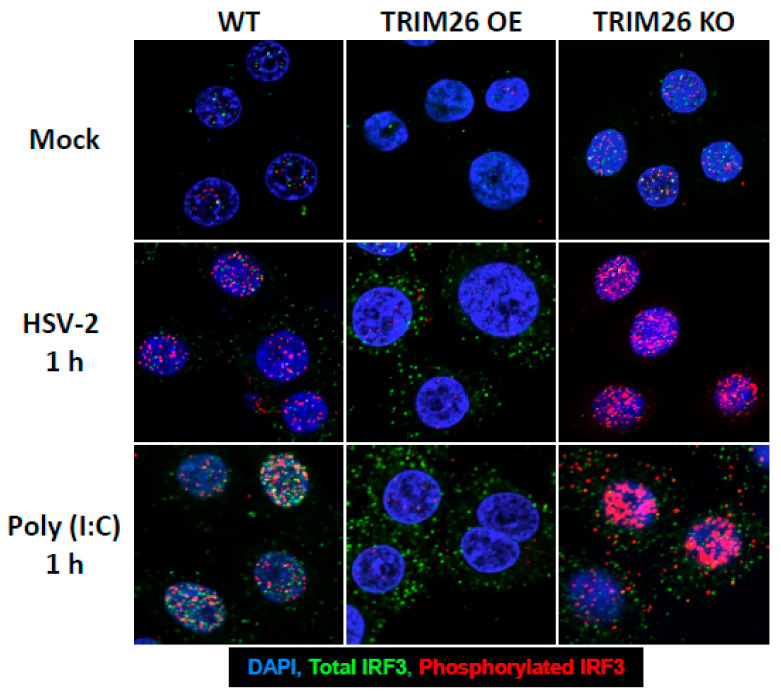
TRIM26 expression negatively regulates activation of IRF3. WT VK2, TRIM26 OE and TRIM26 KO cells were left uninfected (mock) or infected with HSV-2 or treated with Poly I:C, for 1 h. Cells were then fixed and stained as indicated in the colour guide below: DAPI (blue) total IRF3 (green), or phosphorylated IRF3 (pIRF3) (red). Occasional colocalization stain (yellow) in nuclei indicates IRF3 and pIRF3 are colocalized. Images were captured on confocal microscope Representative images are shown. Magnification ×2520.

**Figure 6 viruses-13-00070-f006:**
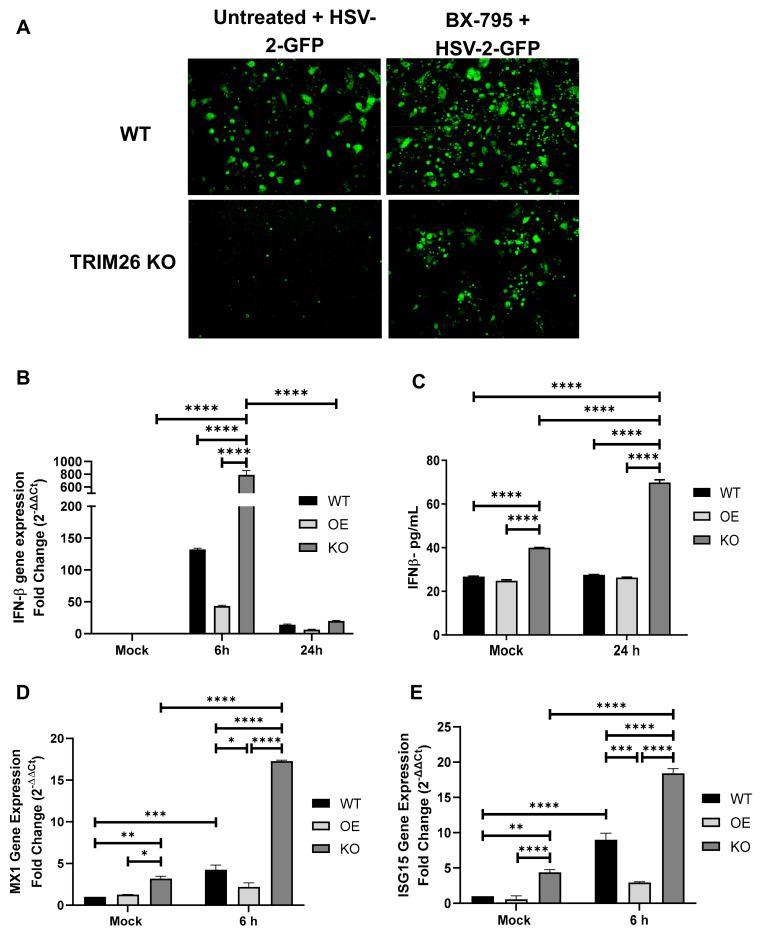
TRIM26 depletion significantly enhances IFN-β and interferon stimulated genes (ISGs) through IRF3 pathway during HSV-2 infection. (**A**) WT VK2 and TRIM26 KO cell lines were pretreated with BX795 (inhibitor of IRF3 signaling pathway) or left untreated for 1 h and then infected with HSV-2-GFP (MOI = 1). At 16 h post HSV-2 infection, cells were visualized by fluorescence microscopy, and images captured using an inverted immunofluorescent microscope (EVOS). Magnification x20. Representative images are shown from *n* = 4 experiments. (**B**) RNA was extracted from WT VK2 (WT), TRIM26 OE (OE) and TRIM26 KO (KO) cell lines after 6 h and 24 h post HSV-2 infection and subjected to qPCR to detect INF--β RNA (*n* = 3), with statistical significance **** *p* < 0.0001. (**C**) WT, OE and KO cell lines were either uninfected (mock) or infected with HSV-2 (MOI = 1) for 24 h, supernatants collected for ELISA measurement of secreted IFN-β. Data shown are mean ± SEM (*n* = 3) with statistical significance **** *p* < 0.0001. (**D** and **E**) RNA was extracted from WT, OE and KO cell lines after 6 h of infection or from mock infection controls and subjected to RT qPCR using specific primers for two ISGs: MX1 and ISG15. Fold changes relative to uninfected WT controls are shown. Data represent mean ± SEM (*n* = 5). Statistical significance: (**D**) **** *p* < 0.0001, *** *p* = 0.0009, ** *p* = 0.002, Mock KO vs. Mock OE * *p* = 0.02 and HSV-2 6h WT vs. HSV-2 6h OE * *p* = 0.01; (**E**) **** *p* < 0.0001, *** *p* = 0.0004 and ** *p* = 0.01.

**Figure 7 viruses-13-00070-f007:**
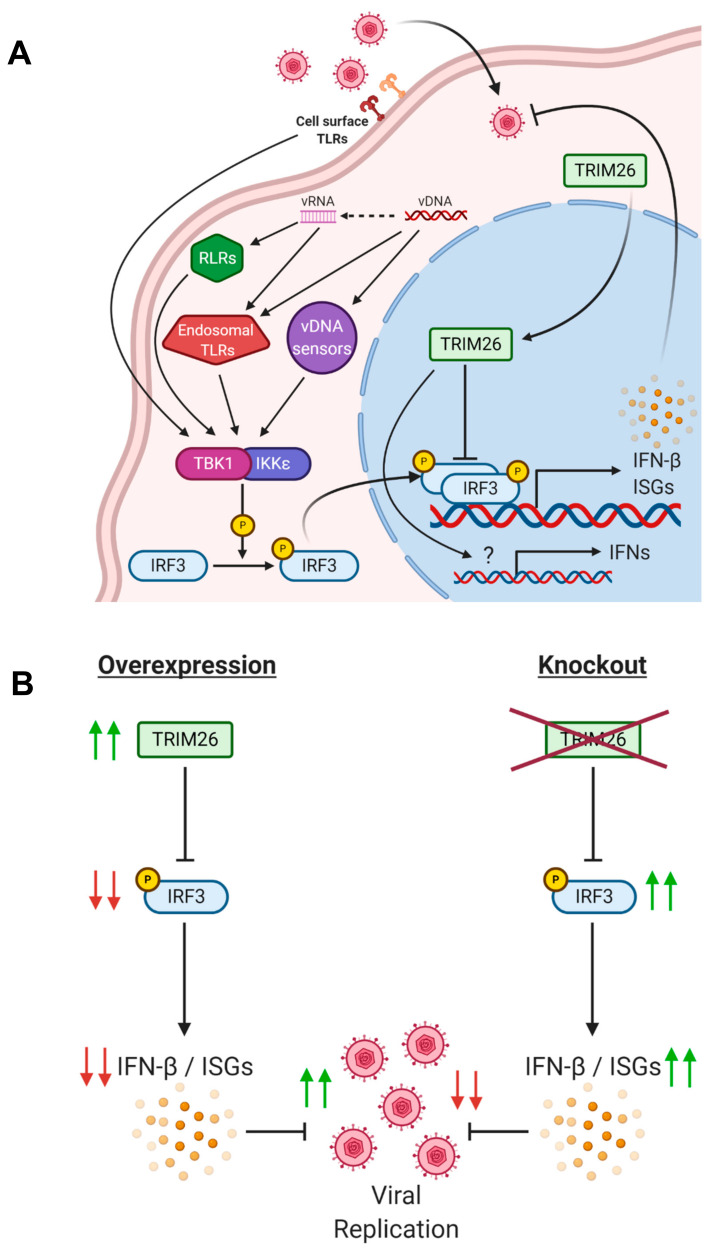
(**A**) Proposed signaling pathways regulated by HSV-2 infection and TRIM26 activation. IRF3 can be phosphorylated, and thereby activated by the TBK1- IKKε complex that is induced by signals from: 1) cell surface TLRs (TLR2 and TLR4); 2) viral dsDNA triggering endosomal TLRs (TLR3, TLR7 and TLR9) or viral DNA sensors (cGAS and IFI16); and 3) viral RNA receptors RLRs (RIG-I and MDA-5). The resulting phosphorylation of IRF3 leads to its dimerization and translocation to the nucleus where it participates in the transcription of IFN-β and ISGs. Viral infection also results in nuclear translocation of TRIM26 that negatively regulates IRF3 activity, resulting in a downregulation of antiviral response. TRIM26 impacts the expression of other interferons after infection through an unknown mechanism. (**B**) Overexpression of TRIM26 downregulates IRF3 expression and its downstream antiviral factors IFN-β and the ISGs MX1 and ISG15, thus leading to increase in viral replication. Knocking out of TRIM26 allows for increased IRF3 expression, thereby stimulating IFN-β and ISG production to inhibit HSV-2 replication. Abbreviations: TLR, toll-like receptor; RLRs, RIG-I-like receptors; IRF, interferon regulatory factor; cGAS, cyclic GMP-AMP synthase; IFI, interferon-inducible; MDA, melanoma differentiation-associated gene; RIG, retinoic acid inducible gene; IFN, interferon; ISG, interferon-stimulated gene.

## Data Availability

Data is contained within the article or Appendix A.

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
