# Peer review of "TRIM26 Facilitates HSV-2 Infection by Downregulating Antiviral Responses through the IRF3 Pathway"

_viruses, 2021, doi:10.3390/v13010070_

Round 1
Reviewer 1 Report
The study performed by Dhawan et al., is highly interesting and important in the field of reproductive immunology. The study describes the mechanistic role of TRIM 26 used by HSV-2 to evade anti-viral responses in vaginal epithelial cell lines to increase viral replication.
The current manuscript is straightforward and verywell written.
Comment which will help to improve the manuscript.
- Given that previous studies (Lines 470-488) have shown that other TRIMs including TRIM5α and TRIM43 play a central role in restricting HSV-1 and HSV-2 infection, can the authors be certain that experiments described in this manuscript with TRIM5α and TRIM43 would not yield the same results seen withTRIM126?
- the authors should explain why they did not use primary vaginal epithelial cells given the known high transcription of TRIM26 in primary vaginal cells, given the fact that they are one of the leading laboratories that routinely work with primary cells from the human reproductive tract.
- It is suggested that additional information be included so that readers will have a better understanding of the cell lines used. For example, The VK2/E6E7 (ATCC CRL-2616) cell line was established in 1996 from the normal vaginal mucosal tissue taken from a premenopausal woman undergoing anterior-posterior vaginal repair surgery. Cells at passage 3 were immortalized by transduction with the retroviral vector LXSN-16E6E7 in the presence of polybrene.
- HSV has been reported to attach to cells by multiple methods. Recent studies have indicated that HSV infection, particularly HSV-2, may also enter cells by mechanism(s) involving endocytosis and/or a phagocytosis-like uptake. A concern with these studies is that differences in viral shedding etc. may be due to effects of TRIM26 on initial uptake in addition to those reported.
- Similar to concerns about uptake, have the authors examined the antimicrobial/ chemokine profiles of VK2 cell secretion before and following transfection to eliminate the possibility that viral killing may explain differences in viral replication and/or shedding?
- Lines 375 and 376 and at other places in this manuscript, it states that “TRIM26 KO show significantly higher localization and translocation of pIRF3 to the nucleus compared to the WT VK2 cells,…”. Findings in these studies do not demonstrate translocation, rather it is suggested. Perhaps the word accumulation would be more appropriate.
- Figure 6A could to be better labeled to indicate that 6A indicates HSV-2-GFP.
Many studies have been performed investigating how HSV-2 suppresses IFN signaling beyond the TRIM proteins (see below for examples). As the authors know, it is a very complicated system, and some effort to integrate these studies into their discussion would be worthwhile.
- Guan et al. (https://www.frontiersin.org/articles/10.3389/fimmu.2019.00290/full) demonstrated that HSV2 can block IRF3 activation and inhibit IFNbupregulation in epithelial cells via the HSV-2 ICP27 protein. Zhang et al. (https://www.jimmunol.org/content/194/7/3102.short) have shown that the HSV-2 ICP22 protein also suppresses IRF3 activation and thus IFNb These manuscripts should be referenced and discussed by the authors. Furthermore, how do the results from Guan et al and Zhang et al. fit in with their model in Figure 7. While Guan et al show that ICP27 binds directly to IRF3, could ICP27 also upregulate TRIM26 as a secondary mechanism? There seem to be two aspects to the suppression of IFNb– one via manipulation of expression of host cell machinery as the authors demonstrate, and one via direct interactions between viral proteins and host cell proteins.
- Previous studies have shown that HSV-2 infection increases the risk of successful HIV transmission. How do the authors results fit into this paradigm? Could the blockade of IFNbupregulation by HSV-2 via TRIM26 inhibit early epithelial innate immune responses crucial for protection? This may be worth a paragraph in the discussion.
- The authors postulate that IFN-independent upregulation of ISGs may account for the earlier upregulation of Mx1 and ISG15 at 6hr compared to the upregulation of IFNbat 24hr. However, an alternate explanation could be the presence of other Type I IFNs such as the multiple subtypes of IFNa. Indeed, given the structure of the Type I IFN locus, it is unlikely to only have IFNbbeing upregulated upon HSV-2 infection. Additionally, IFN lambda 1-3 and IL-27 (both of which have been shown to inhibit HSV-2 infection) can upregulate Mx1 and ISG15. Did the authors examine the expression of these genes at 6 or 24hr? Could the authors postulate whether these antiviral cytokines may have a role to play in their model system?
Author Response
please see attachement

Reviewer 2 Report
The authors demonstrate that a host protein, TRIM26, is a facilitator of HSV-2 infection in VK2 cells. The manuscript is interesting and for the most part, describes a novel observation. The authors also propose that TRIM26 somehow decreases the nuclear localization of IRF3, which is in turn responsible for the attenuation of antiviral responses against HSV-2. While the authors made use of new transgenic cell lines, more proof is needed to verify the protein levels, especially since the cell lines are reported for the first time. Several issues mentioned below will need to be addressed before the authors’ claim can be considered scientifically valid. Introduction: Please cite a comprehensive and recent review article covering all aspects of HSV-2 genital infection (e.g. PMID: 28357380). Line 184: Please mention the actual strain of HSV-2 used. Fig. 1B, the TRIM26 staining is perinuclear and nuclear, why? Please show a higher magnification image. 1MOI infection by HSV-2 in most cases is cytolytic. What happens when a productive infection is induced at a lower MOI? Is the upregulation HSV-2 strain specific? Fig. 2B, Please show TRIM26 only images as well. Staining is more cytoplasmic here. Western data should be included to show (and confirm) side-by-side comparison of protein expression in the two cell types. Fig. 3: Western data that confirms the knockout be shown. Fig. 4, Plaque size vs. plaque count data should be shown. Fig. 5, the results should also be examined at a later time point (e.g. 24h). Fig. 7, please acknowledge that the model is oversimplified.
Round 2
Reviewer 2 Report
Authors have done an excellent job of revising the manuscript. Major results are scientifically validated.